# SHOTSIGHT: EXPLAINING KGE MODELS WITH AN LLM-READY, EXAMPLE-BASED HEURISTIC

## ABSTRACT

This article tackles the critical challenge of explainability in Knowledge Graph Embedding (KGE) models. We introduce a novel case-based reasoning approach called ShotSight, that leverages the latent space representation of nodes and edges in a knowledge graph to generate compelling, human-understandable, example-based explanations for link predictions. By analyzing the impact of identified triples on model performance, we demonstrate the effectiveness of our approach in generating explanations compared to random baselines. We evaluate our method on two publicly available datasets and show its superiority in terms of explanatory power for KGE models. Furthermore, we demonstrate the broader applicability of this technique, extending beyond traditional KGE explanations. Specifically, our method can serve as a valuable aid in constructing relevant "shots" for few-shot prompting within Large Language Models (LLMs) and can be integrated into graph-based Retrieval-Augmented Generation (RAG) systems, effectively making KGE models LLM-ready.

## 1 INTRODUCTION

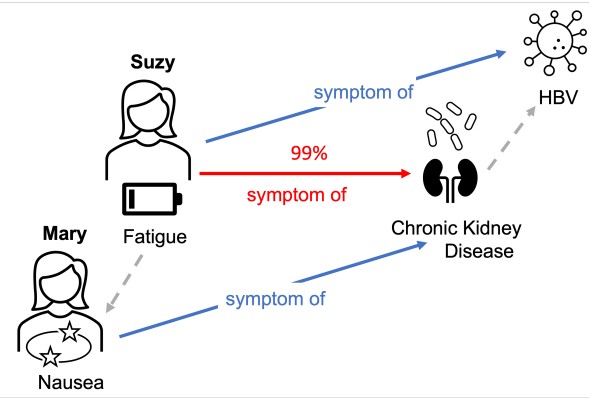

Figure 1: To support prediction of the target statement we identify influential examples by probing the knowledge base constrained w.r.t. the latent-space. This example is drawn from the Fb15k-237 dataset. Predicted plausability score was 99%, and two most influential examples were retrieved as an explanation with the following ranks: 1st: $Nausea \rightarrow symptomOf \rightarrow ChronicKidneyDisease$, 2nd: $Fatigue \rightarrow symptomOf \rightarrow HBV$
.

Link prediction is a common task in knowledge graphs, and often tackled with knowledge graph embedding models, such as ComplEx, DistMult, TransE Trouillon et al. (2016); Yang et al. (2015); Bordes et al. (2013), and others. However, these models lack direct interpretability Bianchi et al. (2020), which is crucial for applications in critical domains like drug discovery and medicine Costabello et al. (2020). Existing explainability methods for KGE models Lawrence et al. (2020); Bhardwaj et al. (2021); Kang et al. (2019); Betz et al. (2022); Baltatzis & Costabello (2024) are limited, and their evaluation approaches and datasets vary. Moreover, the human readability of explanations

is often overlooked. In this work, we propose a new method that generates explanations for link predictions in KGE models using influential examples. We also introduce a dataset for evaluating explanations through user studies. We grounded our approach in the Case-based reasoning (CBR), a problem-solving approach that uses past experiences to inform solutions for new problems, Pradeep et al. (2025) discusses combining CBR and XAI in a broad terms but for the first time it is used for graph and specifically KGE models.

The need for AI systems to be explainable is growing due to legal requirements proposed in response to the increasing risks posed by AI. Specifically, Article 13 of the EU AI Act, titled "Transparency and Provision of Information to Deployers," which will come into force in August 2026, requires that the operation of high-risk AI systems be transparent. Recital 27 clarifies that transparency means allowing traceability and explainability EU (2024), suggesting that methods of explainable AI (XAI) can help achieve compliance.

To address the lack of interpretability in KGE models, highly utilised in biomedical applications, we propose post-hoc interpretability methods inspired by those used for other machine learning models. Our goal is to provide explanations that link predictions back to the original graph, highlighting the links and nodes that contribute the most to a given prediction. These explanations should be understandable to users and provided quickly and efficiently.

However, achieving explainability in KGE models presents several challenges. First, there is a lack of evaluation protocols, metrics, and benchmark datasets specifically designed for assessing the explainability of these models. Existing benchmark datasets, such as FB15k-237 and WN18RR, YAGO3-10 often lack human-readable labels and are not primarily intended for evaluating explainability. We can see this in e.g. FB15k-237 derivative of a Freebase database that was discontinued. One example of triple could be the following: /m/08966, /travel/travel_destination/climate./travel/travel_destination_ monthly_climate/month, /m/05lf_. Not only ids of entities are unknown but also predicates structure is cumbersome and not easily understandable. Similar situation happens also in the case of WN18RR dataset: 06845599, _member_of_domain_usage, 03754979. Despite their lack of interpretability in the sense of understanding what each triple means these benchmark datasets are useful and help establish common grounds for the community. Some newer datasets are more interpretable but not as widely used as ones mentioned before, e.g. CoDEx Safavi & Koutra (2020) comes not only with understandable labels, descriptions and sources but also with multiple languages and is derived from Wikidata and Wikipedia which are actively in usage.

Another challenge for explainability arises from lack of ground-truth explanations apart from synthetic datasets and limited user studies on explanations in the existing literature. Last but not least the model's predictions are ranked based and are not calibrated to represent probabilities directly, which makes it difficult to interpret the results quickly Tabacof & Costabello (2020).

At the moment, there is no way to directly understand what contributed to a prediction of a KGE model. One way to tackle this issue is to use special interpretability methods that work post-hoc. Another way is to design new approaches, e.g., inspired by the methods available for explaining other machine learning models Guidotti et al. (2018). An example of desirable output would be such an explanation that links the prediction back to the original graph pointing out to links and nodes that contributed the most to the given prediction and which removal would result in a decreased probability of the prediction, being in the same time understandable to users. It is also desirable that we can obtain the explanations fast and that they are also memory efficient. Given the above requirements we consulted current literature on the subject to find whether such method exists.

## 1.1 RELATED WORK

The most basic way to identify influential triples would be to perform a simple search over all possible triples that could be removed from the dataset and perform retraining after each such modification of the dataset. This approach is very inefficient as it requires many retrainings of the model. For example, if explanation size, we are interested in, is equal to $|e| = 1$ we need $n$ retrainings of the model for each triple, when $n$ is the number of triples in the training dataset. The number of retrainings is increasing if we allow the explanation to be greater than 1, $|e| > 1$.

This section contains related work and up-to-date SOTA. It starts with brief introduction. Following, each work is described with differentiation factor for ShotSight in mind. Finally, comparable dimensions across presented works conclude the section. Basic principle of the majority of explanation methods presented below is as follows: they try to identify such existing links in the graph that their removal will strongly decrease the probability of the predicted link (this holds for ExplaiNE Kang et al. (2019) and GNNExplainer Ying et al. (2019) but not for GraphLIME Huang et al. (2020)).Worth mentioning is that none of these studies conducted user-studies on effectiveness of human-readability of provided explanations or even whether the list of links/subgraph is enough to constitute the explanation.

Apart from work on explainability aspects of Knowledge Graph Embedding models we would like to bring attention to a similar but seemingly different subject of robustness and adversarial attack approaches for Knowledge Graph Embedding models. Bhardwaj et al. (2021) explored methods of poisoning KGEs with relation inference patterns, which aims at targeting influential triples and design attacks based on it. Another work by Betz et al. (2022) introduced adversarial explanations where they identify regularities in the knowledge graph and plan attacks based on them.

In Pezeshkpour et al. (2019), authors investigated robustness of knowledge graph embedding models with regards to removal or addition of an influential triple to the training set.

Probably, the most notable, published work on explaining Knowledge Graph Embedding models is GradientRollback (GR) Lawrence et al. (2020) and recent KGEx Baltatzis & Costabello (2024). Both are methods that explain specifically knowledge graph embedding models for link predictions. GR works by storing gradient updates in a separate influence matrix per every training example $t$ (during training) and also per every unique entity and relation in a triple. It then refers to this gradient update matrix during the explanation phase. The influence updates regarding the training triple ($t$) are subtracted from the parameters matrix to obtain a new parameter matrix that simulates the situations of retraining the model without $t$. KGEx applies subgraph sampling and knowledge distillation to train local surrogate models, then uses Monte Carlo ranking to identify which training triples are most important for specific predictions. Neither of these methods present an integration with language models that could be utilized in few shot prompting.

## 1.2 RESEARCH QUESTION

By considering the related work and requirements that we formulated in the above section, we posed the following research question:

**RQ:** *How to provide pertinent explanations for relational learning models trained on large knowledge graphs with reasonable time/memory constraints?*

**In this work, we contribute the following:**

- ShotSight, a novel heuristics for generating explanation graphs and a time-efficient batch mode for generating influential examples.
- An evaluation protocol that includes a novel XAI test set for assessing the interpretability of explanations from the Fb15k-237 benchmark dataset.
- Availability of code, dataset, and generated explanations as part of the AmpliGraph Python library.

|           | Fb15k-237 | WN18RR | XAI-Fb15k-237 |
|-----------|-----------|--------|---------------|
| Train     | 272,115   | 86,835 | -             |
| Test      | 20,466    | 3,134  | 239           |
| Valid     | 17,535    | 3,034  | -             |
| Entities  | 14,541    | 40,943 | 445           |
| Relations | 237       | 11     | 91            |

Table 1: Details of the two benchmark datasets Fb15k-237 and WN18RR utilised to evaluate performance of the explainability approach and a novel explainability testset prepared within the scope of this work: XAI-Fb15k-237.

In this section we introduce the intuition behind the proposed approach, notation and the concept of explanation graph. We then move to the steps required to obtain explanation graph and speed-up approach to obtain influential examples. Next we present the evaluation protocol employed in this work and metrics for measuring the performance of the explainability approach presented. As an addition we will present an introduction to a small test set we have prepared for testing XAI methods with the goal of reusing it for future user studies.

### 1.2.1 INTUITION

We propose ShotSight, a post-hoc, local explanation approach that explains Knowledge Graph Embedding predictions. Our approach is based on the assumption that to explain why a certain link between two entities is predicted as plausible, we have to look at the latent space representation of that triple (individually at its subject, object, and predicate embeddings) and try to reverse-engineer the training samples that the pattern was extracted from. We attempt to make an educated guess based on the constraint latent space on which triples the target triple was modeled after.

Considering multi-hop neighbouring triples (especially the 1st-hop neighbourhood) is assumed as this is the most likely source of influence when we look at the training of knowledge graph embedding model, which updates embeddings of triple's elements by considering triple level and corrupting either its subject or object. By triple level training we can observe a ripple effect of influence from one triple to another.

### 1.2.2 NOTATION

Let us introduce key concepts and the notation used throughout the article. Let $\mathcal{G}$ be a knowledge graph, denoted as $\mathcal{G} = (\mathcal{E}, \mathcal{R}, \mathcal{T})$, where $\mathcal{E}$ is a set of entities in the graph, $\mathcal{R}$ is a set of predicates in the graph, and finally $\mathcal{T}$ is a set of statements - triples defining specific links between entities $\mathcal{E}$ with types of relations $\mathcal{R}$, e.g.: triple $t_{(s,p,o)} \in T$ represents a directed edge in the knowledge graph $\mathcal{G}$, where $s$ is the head entity (subject), $p$ is the relation (predicate), and $o$ is the tail entity (object). Let $e$ be an entity in $\mathcal{G}$. The 1-hop neighborhood of entity $e$, denoted as $N(e, 1)$, is defined as: $N(e, 1) = \{(s, p, o) : (s, p, o) \in \mathcal{G}, e \in \{s, o\}\}$, it contains such triples in graph $\mathcal{G}$ that either their subject or their object is the same as the entity $e$ for which the neighbourhood is being derived. Consequently we will define an n-hop neighbourhood of an entity, denoted as $N(e, n)$ as: $N(e, n) = \{(s, p, o) : (s, p, o) \in \mathcal{G}; s, o \in S' \cup O'\} \cup \{N(e, n-1)\}$, where $S' = \{s : (s, p, o) \in N(e, n-1)\}$ and $O' = \{o : (s, p, o) \in N(e, n-1)\}$. It contains triples from the $n-1$ neighbourhood and triples that are connected. Building on top of this formalisation we will define a 1-hop neighbourhood of a triple $t_{(s,p,o)}$ as: $N(t_{(s,p,o)}, 1) = \{N(s, 1) \cup N(o, 1) \setminus T\}$ and consequently we will define n-hop neighbourhood of a triple $t_{(s,p,o)}$ as:

$$N(t_{(s,p,o)}, n) = \{N(t, n-1) \cup \{(s, p, o) : (s, p, o) \in \mathcal{G}; s, o \in S' \cup O'\}\} \qquad (1)$$

where $S' = \{s : (s, p, o) \in N(t_{s,p,o}, n-1)\}$ and $O' = \{o : (s, p, o) \in N(t_{s,p,o}, n-1)\}$.

### 1.2.3 SHOTSIGHT ALGORITHM:

ShotSight is an example-based heuristics that consists of four steps: sampling, filtering for examples, aggregating for prototype and assembling the Explanation Graph.

### 1.2.4 PREREQUISITES:

Calibrated Knowledge Graph Embedding model, returning probability estimates as predictions, e.g., following Tabacof & Costabello (2020), in this way we are ensuring that the predictions are bounded and are as close to the real probabilities as the current SOTA allows.

1. **Latent space sampling:** We include sampling step to derive similar entities to the elements of the target triple in order to construct potential example triples that are a corner stone of the method.

   Let's define an ordered set $S^m = \{s_1, s_2, ..., s_m\}$, $S^m \in S$, where elements of the set are entities with the same ordering as in set $S$, as described below. Given, $s$ is a subject of the

target triple, and $f$ is a method of KGE model to obtain embedding of an element; and $s_i$ is another entity in $\mathcal{G}$ different than $s$, and a distance measure: $dist(\mathbf{s}, \mathbf{s}')$ we can define an ordered set as follows:

$$S = \{s_i : dist(f(s), f(s_{i-1})) \leq dist(f(s), f(s_i)) \forall s_i \in \mathcal{E}\} \tag{2}$$

We will also define an ordered set with distances between subject entity and other entities as below:

$$D_S = \{d_i : d_i = dist(f(s), f(s_{i-1})) \leq dist(f(s), f(s_i)) \forall s_i \in \mathcal{E}\} \tag{3}$$

We will now repeat the same operation for the object $o$ of the target triple to obtain set $O^m = \{o_1, o_2, ..., o_m\}$, $O^m \in O$, analogically to the target triple subject we will define ordering for the object entities as follows:

$$O = \{o_i : dist(f(o), f(o_{i-1})) \leq dist(f(o), f(o_i)) \forall o_i \in \mathcal{E}\} \tag{4}$$

Similarly we will also save distances to the object entity for the other entities in set $D_O$ as below:

$$D_O = \{d_i : d_i = dist(f(o), f(o_{i-1})) \leq dist(f(o), f(o_i)) \forall o_i \in \mathcal{E}\} \tag{5}$$

2. **Filtering for example triples:** Filtering step is necessary to assure that the example triples all exists in the training knowledge graph (represent past cases) and therefore were likely to influence the embeddings of the target triple elements. This step is to obtain the Cartesian product of sets $S^m$ and $O^m$ to create a set of candidate triples with the target triple predicate $p$, as denoted below:

$$eG_t = \{S^m \times O^m : (s_i, p, o_i) \in \mathcal{G}\} \tag{6}$$

3. **Aggregating for prototype:** Aggregating - this steps provides a prototype of all the example triples that were identified. This is motivated by an effective in machine learning technique from classification task of prototype method (e.g. as defined Hastie et al. (2009)). The intuition is to find prototypical features of example triples that will be representative for the identified set and could be used instead of the whole set.

Obtain N-hop neighborhoods according to Equation 1 of Example Triples obtained in Equation 6 into a prototype graph $pG_t$ following strict or permissive strategy.

1. Strict - takes the intersection of sets of triples between n-hop neighbourhoods of examples and target triple, as denoted below:

$$pG_t = \cap_{i=1}^{len(eGT_t)} N(t_{(s,p,o)_i}, n) \cap N(t_{(s,p,o)}, n) \tag{7}$$

2. Permissive - takes the union of sets of triples between n-hop neighbourhoods of examples and intersect it with the target triple n-hood neighbourhood, as denoted below:

$$pG_t = \cup_{i=1}^{len(\mathcal{T})} N(t_{(s,p,o)_i}, n) \cap N(t_{(s,p,o)}, n) \tag{8}$$

4. **Assembling the Explanation Graph:** the last step combines the results of the previous steps into an Explanation Graph $EG$. Assembling joins all the previous steps together and is mostly used for the visualization of the explanation derived in the context.

$$EG = pG_t \cup eG_t \cup \{t\} \tag{9}$$

## 1.3 EVALUATION PROTOCOL

We evaluated our explanation approach by generating influential examples (with associated scores) for TransE models on Fb15k-237 and WN18RR and for the ComplEx model on FB15k-237. For each setup, we compared our proposed method to a baseline, modified the training data based on the explanations, and then retrained the models. Our evaluation focused specifically on the generation of influential training examples.

To evaluate the method we posed the below hypotheses, that we put to test in experiments:

H1 Probabilities are highly correlated before and after retraining with the explanation removed. Moreover, the slope coefficient is $< 1$, meaning that the original probabilities are higher than respective probabilities after retraining.

H2 Plausibility of triples are the lower the more explaining triples are removed with magnitude of a drop reflecting the rank of triples importance.

H3 Retrained model scores target triple as less plausible (compared to the original model scoring the same target triple) upon removal of the explanation.

### 1.3.1 BASELINE:

As a baseline, we used a constrained random explanation approach that selects triples sharing the same predicate as the target triple, regardless of their connection to it. Some other baselines could have been utilised like random triple from the neighbourhood of the target triple. Our baseline increased task difficulty because we focus on explanations with the same predicates, which limits the available pool of triples for explanations to this subset. We can denote the baseline in a mathematical formulation as follows. First, we define a set $E$ of all triples that have the same predicate as target triple $t$:

$$E = \{(e', p', e'') : (e', p', e'') \in \mathcal{G}, p' = p\}$$

Then, we draw a random sample of $n$ triples from set $E$ as a baseline explanation, equal in a size to the number of triples obtained from our heuristics.

$$E_1, E_2, ..., E_n \overset{\text{iid}}{\sim} N(0, \sigma_Z^2)$$

### 1.3.2 METRICS

In this work we used a metric called probability difference measured as percentage. It is similar to the metric used in Lawrence et al. (2020) with the difference that $PD$ used in this work takes into account both explanations that increase the score after retraining and the ones that decrease the triple scores after retraining. It is used, to measure the difference between prediction scores obtained by originally trained KGE model on a target triple and a prediction obtained from the model trained on the dataset without explanation of such training triple, it is defined as follows:

$$PD = \frac{(M(t) - M'(t)) * 100}{M(t)} \tag{10}$$

### 1.3.3 EXPERIMENTS:

1. Remove-and-Retrain (ROAR) Hooker et al. (2019): We removed explanation from the dataset and retrained the model (ROAR protocol) on the modified dataset without explanation for two cases removing only the most influential example triple and removing full set of examples returned by the method.

2. Reversed-Remove-and-Retrain (rev-ROAR): We removed all triples with same predicate as Target Triple (set $E$) and instead added only the explanation (influential examples are restricted to be of the same predicate type as a target triple by default). In this scenario we wanted to test whether model can recover from a loss of majority of its influential examples. We also explored two cases leaving only the most influential example triple and leaving the full set of examples returned by the method.

Note: Both models: original and the one retrained in each experiment were trained exactly the same, with SOTA hyperparameters for the TransE and ComplEx models on the dataset with early stopping. This has the following consequence of evaluation time being very long. For the four scenarios presented we had to train 5 models per dataset per target triple per model.

## 2 RESULTS

In this section we present the experimental results we have obtained from the experiments presented in the evaluation section.

We trained the KGE models with the state of the art hyperparameters for respective benchamrk datasets, we obtained the following results. For Fb15k-237 trained with TransE model we got 0.20 H@1, and 0.30 MRR scores. For WN18RR trained with TransE we obtained 0.05 H@1 and 0.22 for MRR scores. For ComplEx for FB15k-237 we got 0.21 Hits@1 and 0.31 MRR. We have also trained other models and generated explanations with our method for them, their performance is reported in the Appendix along with example explanations obtained for them. Table 4 lists training parameters for models used in the experiments.

The H1 hypothesis was confirmed in all the cases for our method and random, scores after retraining were correlated and whenever we removed the explaining triples with same predicate the respective plausibility after retraining was lower, confirming also H3. Above epoch 20 the Pearson correlation coefficient is 1 and predictions are perfectly correlated. To get a better view of how the probability scores fluctuate across training, at every 10th epoch we make predictions for target triples. We observed an interesting property of the model's prediction ability, that we call **recoverability**, we expected the scores to drop after retraining but we didn't expect that the model could recover from the loss of certain data points, see Figure 2a.

We can see how effectiveness of the explanations depends on how long the model had to recover since the longer the training time the lower the difference between probabilities with original model and the one with influential triples removed. In the case of Fb15k-237 we could see that model recovered almost fully to predicting triple as plausible after 80 epochs of training (Figure 2a). This was not the case for WN18RR (see Figure 2b) where single triple could not convey this much information as in the previous dataset.

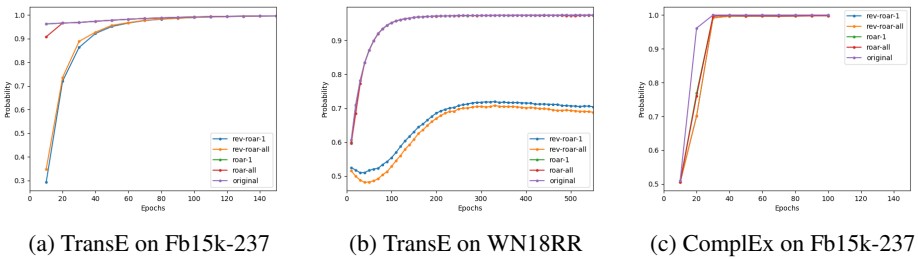

(a) TransE on Fb15k-237     (b) TransE on WN18RR     (c) ComplEx on Fb15k-237

Figure 2: (a) TransE on Fb15k-237. Target triple probability across different epochs. We can see that the probability difference between models before and after explanation removal is the lowest at the beginning of the training. It changes in such a way that model can recover it's predictive ability to predict on a given triple. It means that to make an evaluation of an explainability method, one has to consider the time aspect of the prediction. (b) TransE on WN18RR. Target triple probability across different epochs. (c) ComplEx on Fb15k-237. Target triple probability across different epochs with different experiments.

The H2 hypothesis was also confirmed, we could see that the more triples were removed from training the bigger the difference between probabilities become. The hypotheses allowed us to be confident that our assumptions about impact that the explanations have on the model are correct. Interestingly, longer training reduced these differences, suggesting the model can compensate for missing triples by leveraging other remaining triples—highlighting the graph-based model's ability to "deduce" plausibility and the ripple effects across training.

Appendix Tables B show example explanations generated for three datasets. Table 2 reports average explanation generation times for models trained on different datasets. Our method, ShotSight, explains a single triple in about 5 seconds on average, compared to GradientRollback's 6 minutes for Fb15k-237. Although ShotSight clearly offers faster explanation generation, further investigation is warranted to fully understand the differences between methods. Due to the substantial cost involved, a direct comparison with GradientRollback was not undertaken beyond below assesment.

**Comparison with Other Works**    We explored GR but due to feasibility we couldn't benchmark our method against it and KGEx authors have not released the codebase making comparisons also not feasible. We used available code for GR and compared its results with our proposed ShotSight. For ShotSight it takes 5.09s on average to obtain explanation for the triple in the test set of Fb15k-

237 dataset. 5.85s for the test triples in CODEX-M dataset and 12.33s for WN18RR test triples on average. When comparing with the GradientRollback reported times for Fb15k-237 we can see to obtain an explanation for a single triple it takes 6±7 minute (as reported in their article). Moreover, GR approach requires enabling a special training mode with batch size of 1 making it very slow. It also requires much more memory than the initial dataset size to store training artefacts. E.g. For embedding size of k=100 and training dataset of size 272,115 (Fb15k-237) apart from model parameters during training we have to store around 272115 x 3 x 100 parameters. For Fb15k-237 the size of this extra matrix is equal to 2.4G which is more than 100 times bigger than the sole dataset size of 22.6Mb. This method although time and memory consuming traces parameters in the training leading to probably more accurate results on the expense of the high resources cost. The resources exhaustion makes it prohibitive for usage in a large scale knowledge graphs.

## 2.1 XAI FB15K-237 DATASET CONSTRUCTION

At the moment there are no available and accepted benchmark datasets for human evaluation of knowledge graph link prediction explainability methods. Each paper published up-to-date follows different evaluation approach. It is, therefore, difficult to compare and draw conclusions about which of the methods gives the best results.

Current benchmark datasets in knowledge graph embedding are not user-friendly for laymen, often using encoded entities with incomplete and misaligned mappings. For example, Freebase fb15k-237 has over 20,000 test triples, making human evaluation impractical. Evaluating just 100 triples for understandability takes about 10 minutes per person, translating to roughly 34 hours for the entire test set—requiring multiple annotators for reliable results. This estimate excludes the time needed to assess full explanations, which can span multiple triples or entire subgraphs. Due to the high cost and effort, comprehensive human evaluation is infeasible for all published methods and datasets. This challenge, combined with the lack of standard evaluation frameworks, often limits assessments to synthetic metrics rather than human-grounded usefulness.

Recent publications on explainable link prediction do not provide any evaluation framework that could contribute to fair comparison of explainability approaches, whereas explainability of methods requires strict protocols involving a human-evaluation Doshi-Velez & Kim (2017).

A contribution to rigours evaluation could be a benchmark datasets that would make it possible to compare different explainers of link prediction models.

The following sections describe the desiderata of such dataset and then the evaluation protocol of obtaining such dataset with an example of Fb15k-237 Bordes et al. (2013) dataset which is a benchmark dataset commonly accepted in the community for assessing link prediction models. Sub-setting already existing benchmarking datasets vs creating another dataset from scratch has an advantage of being already familiar to the community and can eliminate the problem of running explantation methods for models that benchmark results are not known. The disadvantages of such approach is preserving biases available in the original datasets, dataset becoming obsolete (e.g. Freebase was discontinued).

**Desiderata** The dataset should prioritize 1) human-readable triples, 2) aligning with layman understanding and 3) minimizing cognitive load for evaluation (limited size). 4) It must also be diverse across predicates, 5) be publicly available, and 6) be based on a benchmarked dataset to mitigate issues arising from model limitations. It should also fulfil more technical properties has no self-relations, no duplicates. To achieve this we propose one that is human filtered.

Such dataset can provide a ground for a fair comparison across different explainability methods and interpretability of their explanations. In Figure 3 we present a small user study we ran in the lab to curate a subset of Fb15k-237 with triples that could be used in such a user study evaluation of the explanations. Similar protocol could be employed for other suitable datasets with more interpretable entities and relations (e.g. CoDEx).

**Dataset Construction** The dataset is a subset of a known benchmark in the knowledge graph embedding community - Fb15k-237, to select triples we utilised a following procedure: we scored all the triple with the model and from among triples with highest score we selected these that were readable for our 3 evaluators. The three evaluators were knowledgable in the knowledge graph and

explainability field. The detailed overview of the dataset is presented in 1.2. There is an intentional bias in a benchmark dataset regarding the triples that score high. It also contains any other bias that was included in the original benchmark dataset Fb15k-237 as it is a subset of it.

### 2.1.1 IMPLEMENTATION:

All experiments were implemented using Python 3.7 with Knowledge Graph Embedding library AmpliGraph version 2.0, using TensorFlow 2.10 . All experiments were run under Ubuntu 18.04 on an Intel Xeon Gold 6142, 64 GB, equipped with a Tesla V100 16GB.

## 3 DISCUSSION

In this work we introduce a novel heuristics to generate explanations for knowledge graph embedding models. It works by generating influential examples from the constrained latent space search. We evaluated the approach with a protocol involving novel XAI test set for evaluating interpretability of explanations for the users. For future work we would like to compare how this approach work on the GNN architectures since it is a model agnostic heuristic. One disadvantage, that we are aware of, is that our approach is a heuristics, we are doing some follow-up research on how to provide estimation guarantees for this approach and if it is even possible. In the same time, we have decided to go forward with the heuristic approach because of immense memory consumption and slow execution time of other methods. Another limitation of ShotSight is its dependence on the quality of past cases and the effectiveness of criticism strategies used in example-based explanations Kim et al. (2016) which we indirectly support in the aggregation step.

Overall, the evaluation results support the effectiveness of our explanation approach based on influential examples. By removing these examples and retraining the model, we observed a decrease in the plausibility of the target triple, indicating the importance of the identified examples in the original prediction. This demonstrates the potential of our method to provide meaningful and informative explanations for link predictions in knowledge graph embedding models.

In addition to the evaluation of our approach, we also introduced a novel XAI test set for evaluating the interpretability of explanations from knowledge graph embedding models. This test set, based on the Fb15k-237 benchmark dataset, provides a valuable resource for future user studies and benchmarking of explainability methods. Furthermore, we will made the code, dataset, and generated explanations available as part of the AmpliGraph Python library, facilitating the adoption and further exploration of our approach by the research community.

The core innovation of ShotSight extends beyond traditional KGE explanation techniques by directly addressing the field of LLM-augmented knowledge retrieval. Our method's ability to generate contextually relevant 'shots' – specifically, representative triples from the knowledge graph; provides a crucial bridge to few-shot prompting within Large Language Models. These generated examples, derived through our heuristic, can condition the LLM's "reasoning" process. It offers a simple way for obtaining knowledge-grounded examples from the knowledge graph databases, something that cannot be done with other explainability methods for KGEs.

In conclusion, our work contributes to addressing the challenge of explainability in knowledge graph embedding models. By leveraging influential examples and generating explanation graphs, we offer a novel approach to explaining link predictions. The evaluation results demonstrate the effectiveness of our method in providing interpretable and informative explanations. We believe that our approach has the potential to enhance the interpretability and trustworthiness of knowledge graph embedding models in various domains, including critical applications such as drug discovery and medicine.

### REPRODUCIBILITY STATEMENT

We provide hyperparameter and other relevant details in the Appendix. We will also release code for method as part of graph machine learning library and code for the experiments to reproduce the results reported in the paper.

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

APPENDIX

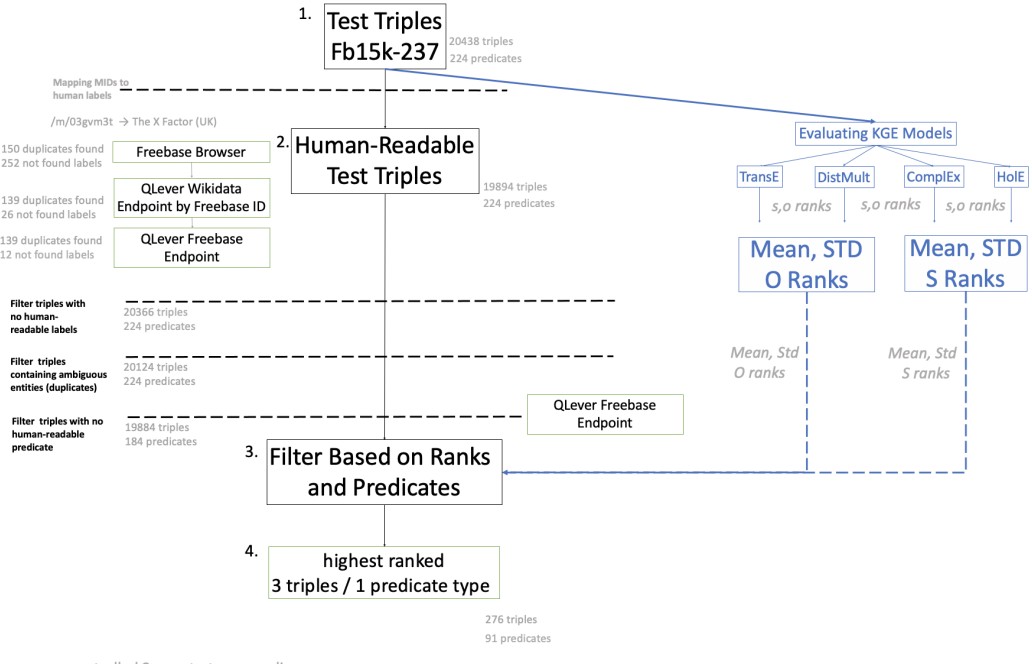

Figure 3: Validation Protocol. The diagram presents steps to be taken to get the dataset for human evaluation. It starts from the original benchmark dataset: Fb15k-237 (box number 1.) and ends at the curated subsets of triples based on properties listed before. The path highlighted in blue refers to the collection of ranks according to different models. The green box specifies selection criteria for constructing the final datasets.

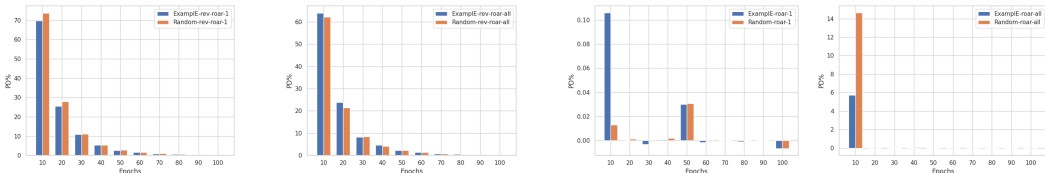

Figure 4: Probability differences between ShotSight and random baseline, TransE trained on Fb15k-237.

## A COMPUTATIONAL COMPLEXITY ANALYSIS

We have analyzed computational complexity of the ShotSight in the scenario of batch explanations. ShotSight requires access to the training dataset so space complexity starts from $O(t)$. Given: $e$ - number of entities (e.g.: 14,541 in Fb15k-237). $k$ - embedding vector dimension (e.g. k=400, TransE on Fb15k-237). $m$ - number of nearest neighbours considered (parameter of ShotSight default m=25). $t$ - number of triples in the train set. $x$ - number of examples, as explanation $x << t$. we can split computational complexity into steps: 1) Sampling - this step is entirely dependent on the nearest neighbour algorithm implementation, in the experiments we used implementation provided in sklearn, which by default tries to adjust parameters for best efficiency. In the worst case scenario it uses a brute force approach which complexity of the prediction time is $O(e \times k \times m)$ with negligible complexity of initialization of the algorithm and negligible space complexity too. In the best case

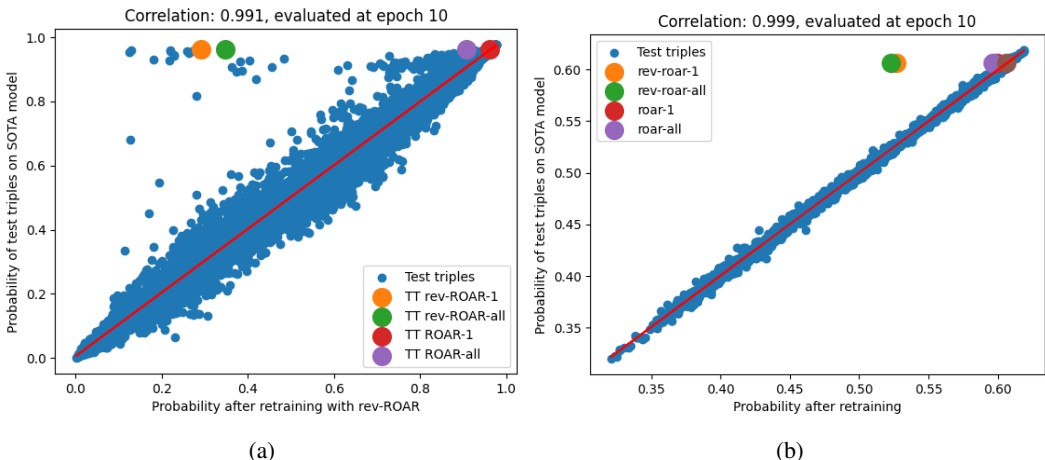

Figure 5: (a) TransE on Fb15k-237 - Probabilities correlation before and after retraining model with reversed ROAR only explanations is left in the training dataset among triples with same predicates as target triple. Above epoch 20 the Pearson correlation coefficient is 1 and predictions are perfectly correlated. (b) TransE on WN18RR - Probabilities correlation before and after retraining model with ROAR.

scenario kNN algorithm tries to adjust the inner data structure for optimized inference time with the cost of initialization and space e.g. in the case of KD-Tree it is $O(k \times e \times log(e))$ of extra initialization time and $O(k \times e)$ space with a benefit of inference time being $O(m \times log(e))$. ShotSight needs to find m nearest neighbours for both subject and object entity (in the default case) in this step. 2) Filtering for example triples - in this step we need to take a cartesian product of obtained sets of neighbours in step 1: which leaves us with $O(m^2)$ (default case, in full case it is $O(m^3)$ if we are considering predicates embedding as well) and forces us to filter examples according to the dataset. First we are mapping it into a tuples ($O(t)$, where $t$ is a number of train triples), than we utilize sets intersections implementation in Python with complexity of $O(min(t, m^2))$. In the post-processing step we compute the score per each example obtained. The computational complexity in batch explain is always dependent on the number of target triples to obtain explanations for.

## B MODELS CALIBRATION

Knowledge Graph Embedding models predictions are uncalibrated - meaning they do not represent probability of the triple but rather a plausability score. Explainability approches presented in this work requires models to be calibrated. Achieving such property with knowledge graph embedding models requires additional post-processing to make sure the returned predictions can be interpreted as probabilities. We calibrate the trained model using procedure described in Tabacof & Costabello (2020) on the validation test. Figures 9 and 10 shows the reliability diagram for uncalibrated and calibrated scores compared with perfectly calibrated reference for two datasets. We can see that uncalibrated scores (in blue) are unevenly distributed and concentrated around the lower part of predicted values, whereas calibrated scores (in orange) although not perfect are distributed much more evenly across the mean predicted values space. By calibrating pre-trained knowledge graph embedding models to return values between 0 and 1 that are more evenly distributed, we return more reliable and interpretable probabilities of triples.

## USE OF LARGE LANGUAGE MODELS

This manuscript benefited from the use of a large language models (LLMs) to assist with language polishing and improving readability. The authors reviewed, edited, and approved all content to ensure accuracy and integrity.

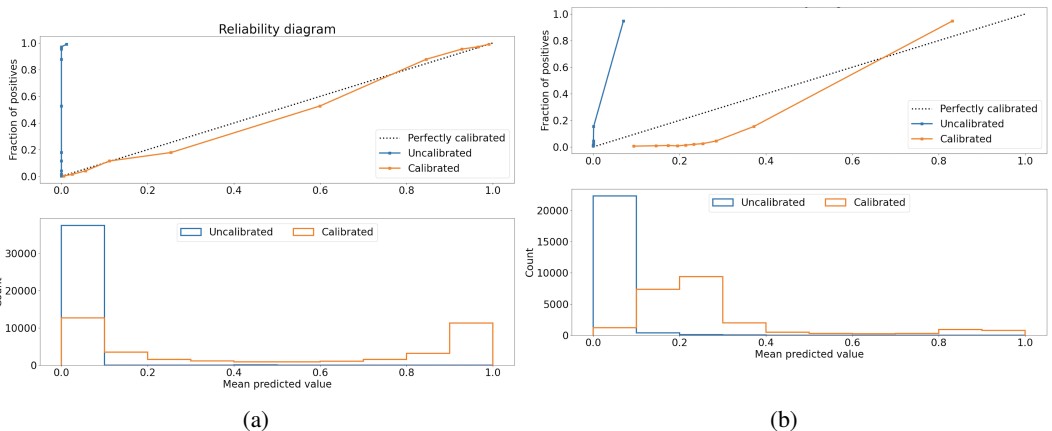

(a)

(b)

Figure 6: (a) Reliability diagram of the TransE model's calibration for FB15k-237 dataset. (b) Reliability diagram of the TransE model's calibration for WN18RR dataset.

| epoch | average | rev-ROAR [%] | | ROAR [%] | |
|---|---|---|---|---|---|
| | | 1 | all | 1 | all |
| 10 ours | -0.507 | 69.694 | 63.971 | **0.106** | 5.717 |
| rand. | -0.478 | 73.757 | 62.186 | 0.013 | **14.647** |
| 20 ours | -0.066 | 25.457 | 23.819 | -0.0 | -0.009 |
| rand. | 0.007 | 27.968 | 21.456 | 0.001 | -0.011 |
| 30 ours | -0.181 | 10.911 | 8.27 | -0.003 | 0.013 |
| rand. | -0.348 | 11.117 | 8.388 | -0.001 | -0.073 |
| 40 ours | 0.009 | 5.312 | 4.735 | 0.0 | 0.034 |
| rand. | 0.096 | 5.435 | 4.154 | 0.002 | 0.048 |
| 50 ours | -0.049 | 2.707 | 2.227 | 0.03 | 0.036 |
| rand. | 0.021 | 2.863 | 2.196 | 0.031 | 0.022 |
| 60 ours | -0.112 | 1.635 | 1.456 | -0.002 | -0.016 |
| rand. | -0.057 | 1.718 | 1.332 | -0.001 | -0.007 |
| 70 ours | -0.135 | 0.894 | 0.786 | -0.0 | -0.011 |
| rand. | -0.072 | 0.94 | 0.721 | -0.0 | -0.015 |
| 80 ours | 0.053 | 0.574 | 0.496 | -0.001 | -0.001 |
| rand. | 0.002 | 0.599 | 0.456 | -0.001 | -0.038 |
| 90 ours | -0.153 | 0.384 | 0.325 | -0.0 | 0.028 |
| rand. | -0.421 | 0.402 | 0.303 | -0.0 | -0.0 |
| 100 ours | -0.153 | 0.23 | **0.203** | -0.007 | -0.01 |
| rand. | -0.03 | **0.244** | 0.176 | -0.007 | -0.005 |

Table 2: TransE on Fb15k-237 - Probability difference between original model and models retrained using two different scenarios ROAR and rev-ROAR considering most influential triple (1) and all triples from the obtained explanation (all). We can see that when retraining the model with only a single triple of given predicate (rev-ROAR-1) the model can recover from it's initial almost 70% probability drop at epoch 10th to a little over 0.2 difference at epoch 100th, we can observe the same pattern but faster for the training with full explanation. On the other hand when we look at the ROAR experiment we can see that removing a single triple has only influence epoch 10th of training with 0.1% probability drop, this is increased when all triples are removed to 6%.

| epoch | average | rev-ROAR [%] | | ROAR [%] | |
|---|---|---|---|---|---|
| | | 1 | all | 1 | all |
| 10 ours | 0.0 | 13.515 | 15.018 | **1.118** | **1.696** |
| rand. | -0.004 | - | - | 0.089 | -0.306 |
| 20 ours | -0.004 | 26.975 | 29.598 | 0.008 | 3.5 |
| rand. | -0.027 | - | 25.585 | -0.021 | -0.058 |
| 30 ours | 0.018 | 34.791 | 37.518 | 0.031 | 1.198 |
| rand. | 0.005 | 34.384 | - | -0.046 | 0.035 |
| 40 ours | -0.004 | 38.778 | 42.265 | 0.048 | 0.041 |
| rand. | 0.02 | - | - | 0.021 | 0.036 |
| 50 ours | 0.006 | 40.661 | 44.588 | 0.038 | 0.095 |
| rand. | 0.025 | 42.159 | 41.134 | -0.01 | 0.004 |
| 60 ours | 0.015 | 42.071 | 45.949 | 0.054 | 0.018 |
| rand. | 0.042 | 43.586 | 42.388 | 0.011 | 0.028 |
| 70 ours | -0.032 | 43.149 | 46.509 | 0.071 | 0.021 |
| rand. | -0.309 | 44.506 | - | 0.005 | -0.002 |
| 80 ours | -0.017 | 42.835 | 46.048 | 0.025 | 0.017 |
| rand. | -0.252 | 44.057 | - | 0.001 | -0.02 |
| 90 ours | 0.047 | 42.512 | 45.655 | 0.01 | 0.012 |
| rand. | 0.058 | 43.58 | - | 0.011 | 0.013 |
| 100 ours | -0.008 | 41.778 | **44.503** | 0.023 | 0.002 |
| rand. | 0.01 | **42.673** | 41.57 | 0.021 | 0.014 |

Table 3: TransE on WN18RR - Probability difference between original model and models retrained using two different scenarios ROAR and rev-ROAR considering most influential triple (1) and all triples from the obtained explanation (all). We can see that when retraining the model with only a single triple of given predicate (rev-ROAR-1) the model cannot recover from initial 14% probability drop at epoch 10th instead it worsen to reach it's peak at around epoch 70th (amounting to 43%) to settle on nearly 42% difference at epoch 100th, we can observe the same pattern but faster for the training with full explanation. On the other hand When we look at the ROAR experiment we can see that removing a single triple has only influence epoch 10th of training with 1.1% probability drop, this is increased when all triples are removed to 1.7%, the difference above epoch 10th is smaller than 1%.

| model/dataset | WN18RR | Fb15k-237 |
|---|---|---|
| TransE | k=350, eta=30 | k=400, eta=30 |

Table 4: Parameters used for model training, trained with early stopping for 4000 epochs using Adam optimizer with lr=0.0001, multiclass-nll loss, seed=0, regularizer L2 with lambda=0.0001

| dataset | total time [s] | time/triple [s] |
|---|---|---|
| FB15k-237 | 104069.3 | 5.09 |
| CODEX | 60364.4 | 5.85 |
| WN18RR | 36048.5 | 12.33 |

Table 5: Time to obtain explanations for triples for respective TransE models trained on different datasets, total time for all triples in the test dataset, time per triple. This also include cases when no explanations where found. We also generated explanations for ComplEx model for FB15k-237 and the respective times were 196682.1 for all test triples and 9.62s per triple - almost twice as much as for a TransE model, which make sense as ComplEx model due to its architecture has twice as much embeddings. For WN18RR the times for ComplEx model where respectively: 38289.0s and 13.09s.

| Approach | TransE | | ComplEx | | DistMult | | ConvE | |
|---|---|---|---|---|---|---|---|---|
| | H@1 | MRR | H@1 | MRR | H@1 | MRR | H@1 | MRR |
| Fb15k-237 | 0.20 | 0.30 | 0.21 | 0.31 | 0.21 | 0.30 | 0.21 | 0.30 |
| WN18RR | 0.05 | 0.22 | 0.47 | 0.50 | 0.43 | 0.47 | 0.44 | 0.47 |

Table 6: MRR and Hits@1 on the Fb15k-237 and WN18RR benchmark datasets.

| step | initialization time | sapce |
|---|---|---|
| $sampling_{KD-Tree}$ | $O(k \times e \times log(e))$ | $O(k \times e)$ |
| $sampling_{brute-force}$ | $O(1)$ | $O(1)$ |

Table 7: Computational time and space complexity of initialization phase.

| step | prediction time | sapce |
|---|---|---|
| $sampling_{KD-Tree}$ | $O(m \times log(e))$ | $O(1)$ |
| $sampling_{brute-force}$ | $O(e \times k \times m)$ | $O(1)$ |
| product | $O(m^2)$ | $O(1)$ |
| mapping | $O(t)$ | $O(t)$ |
| filtering | $O(min(t, m^2))$ | $O(1)$ |
| post-processing | $O(x)$ | $O(1)$ |

Table 8: Computational time and space complexity of prediction phase.

| S | P | O | Score |
|---|---|---|---|
| Billy Idol | languages spoken, written, or signed | English | TT |
| Johnny Marr | languages spoken, written, or signed | English | 0.00075 |
| Chester Bennington | languages spoken, written, or signed | English | 0.00076 |
| Morrissey | languages spoken, written, or signed | English | 0.00077 |
| Loreena McKennitt | languages spoken, written, or signed | English | 0.00077 |
| Gordon Lightfoot | languages spoken, written, or signed | English | 0.00080 |
| Alan Stivell | languages spoken, written, or signed | English | 0.00080 |
| Robert Plant | languages spoken, written, or signed | English | 0.00080 |
| Oleg Skripka | languages spoken, written, or signed | French | 0.00091 |
| Alan Stivell | languages spoken, written, or signed | French | 0.00091 |
| Oleg Skripka | languages spoken, written, or signed | Russian | 0.00097 |
| Oleg Skripka | languages spoken, written, or signed | Ukrainian | 0.00117 |

Table 9: Example explanation for a test triple in CODEX-M dataset - first row represents Target Triple (TT). The lower the score, the closer is example to the Target Triple.

| S | P | O | Score |
|---|---|---|---|
| Artie Lange | /influence/influence_node/influenced_by | Jackie Gleason | TT |
| George Carlin | /influence/influence_node/influenced_by | Danny Kaye | 0.17232 |
| Conan O'Brien (aka Big Red) | /influence/influence_node/influenced_by | Danny Kaye | 0.19481 |
| Conan O'Brien (aka Big Red) | /influence/influence_node/influenced_by | Steve Allen | 0.24502 |
| Bill Maher | /influence/influence_node/influenced_by | Steve Allen | 0.25498 |

Table 10: Example explanation for a test triple in Fb15k-237 dataset - first row represents Target Triple (TT). The lower the score, the closer is example to the Target Triple.

| S | P | O | Score |
|---|---|---|---|
| 02314321 | _hypernym | 08102555 | TT |
| 02314001 | _hypernym | 08102555 | 0.02788 |
| 02313495 | _hypernym | 08102555 | 0.04839 |
| 01928360 | _hypernym | 08102555 | 0.05155 |
| 02314717 | _hypernym | 08102555 | 0.06077 |
| 01928737 | _hypernym | 08102555 | 0.06294 |
| 02321759 | _hypernym | 02316038 | 0.09046 |

Table 11: Example explanation for a test triple in WN18RR dataset - first row represents Target Triple (TT). The lower the score, the closer is example to the Target Triple.

