# OpenReview forum: "ShotSight: Explaining KGE Models with an LLM-Ready, Example-Based Heuristic"
_ICLR.cc/2026/Conference — ICLR 2026 Conference Withdrawn Submission_

### Official Review · Reviewer_Ck6s · 2025-10-25

**Soundness:** 1
**Presentation:** 1
**Contribution:** 1
**Rating:** 2
**Confidence:** 5

**Summary:**

This paper proposes ShotSight, presented as a novel case-based reasoning heuristic for explaining Knowledge Graph Embedding (KGE) link predictions. It identifies potential explanatory triples by finding entities similar to the target triple's subject/object in the KGE latent space, filtering these candidates for existence in the training graph, and aggregating neighborhood information. Evaluation uses a remove-and-retrain protocol (ROAR/rev-ROAR) on FB15k237/WN18RR, comparing against removing random triples with the same predicate. The authors also introduce a small curated dataset and claim ShotSight can provide "shots" for LLM prompting, making KGEs "LLM-ready".

**Strengths:**

1. **Problem scope:**: Attempts to address the important problem of explainability for KGE models. Although I would like to see the same for harder settings, where the new foundation KG models work on.
2. **Speed:** The proposed heuristic is faster than gradient-tracking or influence function methods (lines 375-388), which is an advantage provided it produces meaningful explanations.
3. **Example-based explanations:** Attempts to provide example-based explanations, which are often easier for humans to understand than value-based explanations.

**Weaknesses:**

1. **Heuristic-based:** Relies on an unjustified assumption (shown neither through theory nor results) linking latent space similarity to explanatory influence.
2. **Poor Results Presentation:** Results are presented in a confusing way that obscures direct comparison of the method's effectiveness against the baseline using the defined PD metric.
3. **Inadequate Evaluation:** No comparison with any established KGE explanation method regarding explanation quality/fidelity. The work motivates the need for human-readable explanations, but there are no results to back that the explanations generated are human-readable. This warrants a user study if one targets such a claim. The baseline used is also trivial.
4. **Unsupported Claims:** Major claims about making KGEs "LLM-ready" are entirely speculative and lack any empirical validation. To make such a claim in the abstract, there have to be experiments supporting it.
5. **Limited Scope:** Tested only on TransE/ComplEx and two datasets in the legacy transductive setting. The field has moved far away from this setting in recent days into the regime of fully-inductive and building foundation models for it. Similar studies on these new settings would be more useful.

**Questions:**

1. The core assumption is that latent space neighbors are influential examples. Provide direct evidence for this. How does the influence ranking from ShotSight (based on distance) compare to rankings obtained via established methods like influence functions or gradient-based attributions on the training examples?
2. Why was no comparison made against GradientRollback or other KGE explanation methods in terms of explanation quality (ex., using ROAR or other metrics), even on a small scale, to contextualize ShotSight's performance? Speed is irrelevant if the explanations are poor.
3. Provide concrete experiments to support the claim that ShotSight's outputs are useful as "shots" for LLMs. Show improved few-shot performance using these examples compared to random or other selection strategies.
4. How was the XAI-Fb15k-237 dataset curated? Selecting only high-scoring, expert-readable triples might create a biased subset unrepresentative of typical KGE predictions or challenging explanation scenarios.

---

### Official Review · Reviewer_mQEy · 2025-10-29

**Soundness:** 1
**Presentation:** 1
**Contribution:** 2
**Rating:** 2
**Confidence:** 4

**Summary:**

ShotSight proposes a case-based (example/prototype) heuristic to “explain” KGE link predictions by sampling neighbors in latent space, filtering to existing triples, aggregating to a prototype subgraph, and then validating via remove-and-retrain (ROAR/rev-ROAR). The paper also introduces a small human-readable subset (“XAI-FB15k-237”). Claims include faster explanation generation than GradientRollback and potential LLM few-shot/RAG use.

**Strengths:**

- Timely problem: explainability for KGE models, with an explicit goal of human-readable outputs.
- Clear intuition (example-based/CBR) and a simple four-step pipeline that is easy to re-implement.
- Computationally cheap compared to gradient-tracking methods; rough complexity analysis included.

**Weaknesses:**

- **Limited novelty & unclear added value.** The method is essentially nearest-neighbor retrieval in embedding space plus simple set operations; the “influence” is only *assumed* and then *imputed* via ROAR, overlapping heavily with prior remove-and-retrain/influence-style ideas. There is no theoretical justification or causal identification of influence beyond correlation.
- **Evaluation is inadequate and sometimes misleading.**
  - Main baseline is a weak “random same-predicate” selector; no strong baselines (GR, KGEx, ExplaiNE) are *properly* compared on the same tasks/metrics. Reporting wall-clock claims vs GR from different setups is not a fair comparison.
  - The ROAR and especially **rev-ROAR** (remove *all* same-predicate triples) are artificial and confounded by distribution shift; results are then interpreted as “recoverability,” weakening the explanatory claim.
  - The sole metric (percentage probability difference after calibration) is idiosyncratic, sensitive to calibration noise, and does not evaluate *explanation quality* (fidelity/precision, comprehensibility) or standard XAI desiderata.
  - Reported model accuracy is weak/inconsistent (e.g., TransE on WN18RR Hits@1 = 0.05; later tables show much higher numbers for other models), undermining conclusions drawn from those models.
- **Speculative LLM claims without experiments.** The “LLM-ready” few-shot/RAG positioning is not validated—no LLM tasks, no ablations, no human or automatic measures showing improvements.
- **Calibration dependency and ambiguity.** The approach assumes calibrated KGE probabilities, but calibration details (splits, reliability, stability across relations/entities) are brief and not tied to explanation quality; probability shifts during retraining confound the metric.
- **Formatting and clarity issues.** Numerous typos, broken numbering, stray page counters, inconsistent notation, and figures/tables referenced but not clearly presented; some claims about “state-of-the-art hyperparameters” contradict observed scores.

**Questions:**

No questions.

---

### Official Review · Reviewer_Q9iV · 2025-11-01

**Soundness:** 2
**Presentation:** 2
**Contribution:** 2
**Rating:** 2
**Confidence:** 3

**Summary:**

This paper proposes a cased-based method for explaining knowledge graph embedding models. It leverages the latent space representation of nodes and edges in a KG to generate explanations for link prediction results. Compared to random baseline, the proposed method is effective.

**Strengths:**

Improving the explanability of neural models is important for application and explaining KGE models worth to investigate.

**Weaknesses:**

1. The citation format of the paper is wrong.
2. As introduced in the related work, there are works such as GR and KGEx published for explaning knowledge graph embedding models.  While in the experiments, the proposed method is not compared to existing works to evaluate which explanation from these method is better.
3. It is stated that the proposed method could serve as a valuable aid in constructing relevant "shots" for few-shot prompting within LLMs and can be integrate into RAG system, which is interesting but lack experiments support the statement.

**Questions:**

See above.

---

> ### Author Response · Authors · 2025-11-22
>
> We thank the reviewer for their time in assessing our submission. We would like to reply to the pointed weaknesses and ask for clarification:
>
> 1. Regarding the citation format, we respectfully ask for specific clarification on what is considered incorrect. Our submission uses the official ICLR 2026 LaTeX template, as recommended in the conference guidelines, and applies a consistent citation style throughout the paper. If a particular citation or formatting detail is problematic, further guidance from the reviewer would be appreciated.​
>
> 2. The lack of empirical comparison against GR and KGEx is acknowledged in the article. Implementing GR and KGEx for large-scale experiments presents computational challenges, and to our knowledge, there is no publicly available implementation of KGEx for direct and fair benchmarking. We would welcome advice if the reviewer is aware of accessible resources or efficient alternatives.
>
> 3. The proposed method's capacity to support few-shot prompting and integration within RAG systems is discussed as a potential future direction; direct experimental evaluation is out of scope for this work.

---

### Official Review · Reviewer_5Srx · 2025-11-01

**Soundness:** 1
**Presentation:** 1
**Contribution:** 1
**Rating:** 0
**Confidence:** 5

**Summary:**

This paper presents a method to improve explainability in Knowledge Graph Embedding models.

**Strengths:**

The overall presentation of the paper is poor. It reads more like a coursework report than an academic paper.

**Weaknesses:**

Same as Streghts.

**Questions:**

Same as Streghts.

---

> ### Author Response · Authors · 2025-11-22
>
> We thank the reviewer for their time in assessing our submission. However, we note that the review does not provide any specific, actionable feedback regarding technical correctness, novelty, or empirical validity of our approach. The comments focus solely on a general statement about presentation quality without referencing particular sections, methods, or results.
>
> We respectfully clarify that the paper follows the standard format for ICLR submissions, including an abstract, methodological description with equations, experimental results, and discussion. The work presents a novel method to improve explainability in knowledge graph embeddings, supported by quantitative and qualitative evaluations.
>
> If there are particular aspects of the writing style or structure that appear unclear or inconsistent with expected academic tone, we would be glad to revise those. However, without more detailed guidance, it is difficult to understand the basis for the reviewer’s low scores across soundness, presentation, and contribution.

---

### Note · Authors · 2026-05-02

I have read and agree with the venue's withdrawal policy on behalf of myself and my co-authors.

---

### Meta-Review · Area_Chair_arbV · 2025-12-26

**Summary:**

This paper proposes a case-based method for explaining knowledge graph embedding models, which leverages the latent space representation of nodes and edges to generate explanations for link prediction tasks by sampling neighbors in latent space, filtering to existing triples, aggregating to a prototype subgraph, and then validating via remove-and-retrain (ROAR/rev-ROAR).
The proposed method is effective in comparison with random baselines. The paper also introduces a small human-readable subset.

**Reviewer Concerns:**

The authors only provided rebuttals to two reviewers, which also didn't address their concerns.

**Reviewer Scores:**

The reviewers consistently give negative scores, and the chances for adjusting their ratings are low.

---

### Decision · Program_Chairs · 2026-01-26

Reject